# Prediction and Ranking of Biomarkers Using *multiple* UniReD

**DOI:** 10.3390/ijms231911112

**Published:** 2022-09-21

**Authors:** Ismini Baltsavia, Theodosios Theodosiou, Nikolas Papanikolaou, Georgios A. Pavlopoulos, Grigorios D. Amoutzias, Maria Panagopoulou, Ekaterini Chatzaki, Evangelos Andreakos, Ioannis Iliopoulos

**Affiliations:** 1Department of Basic Sciences, School of Medicine, University of Crete, 71003 Heraklion, Greece; 2Laboratory of Pharmacology, Medical School, Democritus University of Thrace, 68100 Alexandroupolis, Greece; 3EnzyQuest PC, Science and Technology Park of Crete, 100 Nikolaou Plastira Str., Vassilika Vouton, 70013 Heraklion, Greece; 4Institute for Fundamental Biomedical Research, Biomedical Sciences Research Center “Alexander Fleming”, 16672 Vari, Greece; 5Bioinformatics Laboratory, Department of Biochemistry and Biotechnology, University of Thessaly, 41500 Larisa, Greece; 6Institute of Agri-Food and Life Sciences, University Research Centre, Hellenic Mediterranean University, 71410 Heraklion, Greece; 7Laboratory of Immunobiology, Center for Clinical, Experimental Surgery and Translational Research, Biomedical Research Foundation of the Academy of Athens, 11527 Athens, Greece

**Keywords:** biomarker validation and ranking, protein–protein interaction prediction

## Abstract

Protein–protein interactions (PPIs) are of key importance for understanding how cells and organisms function. Thus, in recent decades, many approaches have been developed for the identification and discovery of such interactions. These approaches addressed the problem of PPI identification either by an experimental point of view or by a computational one. Here, we present an updated version of UniReD, a computational prediction tool which takes advantage of biomedical literature aiming to extract documented, already published protein associations and predict undocumented ones. The usefulness of this computational tool has been previously evaluated by experimentally validating predicted interactions and by benchmarking it against public databases of experimentally validated PPIs. In its updated form, UniReD allows the user to provide a list of proteins of known implication in, e.g., a particular disease, as well as another list of proteins that are potentially associated with the proteins of the first list. UniReD then automatically analyzes both lists and ranks the proteins of the second list by their association with the proteins of the first list, thus serving as a potential biomarker discovery/validation tool.

## 1. Introduction

Protein–protein interactions (PPIs) play an important role in the proper function of living organisms. Proteins can interact directly in a binary form or as members of a complex, or can be functionally related without physical interaction (e.g., when they are involved in the same biological pathway). Due to the importance of PPIs in understanding how complex biological networks function, in the past 30 years, high-throughput technologies such as Y2H, protein arrays, co-immunoprecipitation, mass spectrometry and others [1] have been used for the generation and understanding of PPI networks. While these large-scale experimental approaches come with many advantages, they often become time-consuming, are expensive to run and report findings with high false positive rates, low accuracy and low reproducibility. On the other hand, in silico PPI prediction approaches are often faster and of low cost and can be roughly summarized in two categories. In the first category, computational approaches are dedicated to analyzing the genome context and suggest interactions based on information related to structural and evolutionary features of genomes (phylogenetic profiling [2], gene fusion detection [3] and conservation of gene clusters on chromosomes of different species [4]. In the second category, computational methods utilize biomedical literature mining to exploit the information hidden in millions of abstracts/articles stored in various biomedical-related databases such as Medline [1].

Text mining (most particularly biomedical literature mining) methods have been utilized in a variety of challenging problems in biomedical research such as concept discovery in biomedical literature (BioTextQuset+) [5], drug discovery (DrugQuest) [6], disease–gene associations from biomedical abstracts (DISEASES) [7] and biomedical term co-occurrence (CoPub) [8]. The Darling application, for example [9], mines disease-related databases such as OMIM, DisGeNET and Human Phenotype Ontology (HPO). OnTheFly [10] performs named entity recognition in texts and spreadsheets. EXTRACT [11] finds associations between biomedical terms in web pages and consequently the literature.

In the last 20 years, many methods have been developed for PPI extraction from biomedical literature [1]. In particular, PubGene can detect associations between genes/proteins using terms from the MeSH index (Medical Subject Headings) as well as terms from the gene ontology (GO) database [12], HIPPIE provides human PPI networks through PPI network scoring, integration of experimental information and basic graph algorithms [13], iHOP [14] uses genes and proteins as hyperlinks between sentences and abstracts derived from PubMed to provide a protein association network, etc. 

In addition to the aforementioned methods, there are databases/tools such as STRING that integrate information from various sources (literature, experiments, databases and genome context) in order to provide protein associations from a large number of species [15] as well as a large number of manually curated databases that contain experimentally validated PPIs, either extracted from the literature or from repositories of large-scale omics experiments [1] (Papanikolaou et al., 2015) (e.g., IntAct [16], DIP [17], MINT [18]).

UniReD is a computational tool that is used to predict functional associations between proteins based on the Markov cluster algorithm (MCL). UniReD parses the references section of UniProt entries for *Homo sapiens* and *Mus musculus* organisms and by using the “similar articles” feature of PubMed, it creates clusters of functionally associated proteins [19]. UniReD performs quite well after following benchmarks against public databases of experimentally verified PPIs and has been used by experimental biologists to identify known and undocumented interactors of protein(s) of interest [19,20,21]. In addition, UniReD has been used previously [22,23] for the ranking of biomarkers extracted from high-throughput datasets. The whole process was performed manually, and the user had to analyze two lists of genes/proteins, using UniReD results. The first one was a list of proteins/genes known to be associated with the disease/pathway of interest, whereas the second one was a list of proteins/genes that were extracted from high-throughput datasets and had to be validated for relatedness against the first list of genes. The whole process was laborious since the user had to run UniReD for each pair of proteins (see Materials and Methods section for details).

Herein, we present a new powerful feature of the UniReD tool (*multiple* UniReD) which automates the aforementioned process, in an effortless way. Users may provide the two lists of proteins of interest and *multiple* UniReD will return the validated results ranked within a few seconds. This functionality is of particular usefulness in biomarker discovery/validation pipelines, as demonstrated by a test case. UniReD is available at http://bioinformatics.med.uoc.gr/shiny/multiunired/, accessed on 16 September 2022.

## 2. Results and Discussion

We have developed a user friendly and self-explanatory web interface for *multiple* UniReD to assist wet lab biologists with limited to no programming skills.

The multiple search feature of UniReD can be accessed from the main page of UniReD’s web interface and users are redirected to the new dedicated interface for this pipeline. By choosing the analysis button, the user now can view the main interface page of *multiple* UniReD analysis. In order to run the analysis, a two-step procedure has to be followed. Firstly, two separate lists of proteins have to be uploaded, in a comma- or tab-separated format (.csv or .tsv files). The first list (query file) represents proteins whose functional associations the user wants to explore in relation to the second list of proteins (reference file) that are selected by the user and are related to the research topic under investigation (e.g., a specific disease). When uploaded, an overview of these lists is displayed. Then, the organism from which these proteins derive has to be defined by the user. In its present form, UniReD (as well as *multiple* UniReD) can analyze human or mouse proteins but we are planning to expand the analysis to other species as well (Figure 1). 

In order to test the *multiple* UniReD tool, we selected two gene lists that we used in a previous work [22], in which functional associations were searched manually one by one, using UniReD’s web interface. The reference list contains proteins whose implication in breast cancer is well documented in the literature [22], whereas the query list contains proteins that were proposed as biosignatures for breast cancer by a machine learning approach. It should be mentioned that the latter analysis was conducted using the UniProt version of March 2017, whereas for the new analysis, we used the updated version of UniProt (October 2021). This may be a reason why the results of the analysis are slightly different when compared to those of [22], because the two versions of the UniProt database vary and the difference has an impact on UniReD results.

The analysis via *multiple* UniReD is conducted within a few seconds and the results are available in three distinct tables. The first one reports the scores for each pairwise association between proteins of the two lists, accompanied by the overall score for each one of the proteins under investigation. The overall score is reported in the last column, shown in a descending order. In this way, the user can view a ranking of the proteins of interest sorted by their relatedness to the proteins in the reference list. UniProt accession numbers and gene names in parentheses are provided as identifiers for each protein (Figure 2).

The second table describes every pairwise association type for which a score has been assigned. In this table, the first column corresponds to the query proteins whereas the second one holds the reference proteins. In the third column, the user can view the type of association between the query and the reference protein that *multiple* UniReD was able to record. In total, four different cases can be recorded. *InCluster* specifies that both query and reference proteins coexist in a protein cluster of UniReD, i.e., a functional association has been derived from the literature. *Paralogue* states if a paralogue of the query protein has been found in the same cluster with a reference protein. *Orthologue* applies when an association has been found between the orthologues of the query and reference proteins in a different species. In the case where another protein component of a complex that the query protein is part of is associated with a reference protein, it is labeled *Complex* in the third column. The fourth column is used to display the pair of proteins found to be associated. For example, if a paralogue of the query protein has been found to be related to a reference protein, the UniProt accession number and gene name in parentheses of this paralogue are provided. In the case of an orthologue search, the pair of orthologues to the query and reference proteins is shown.

Proteins that are not part of any protein cluster formed by UniReD are excluded from the analysis and become entries of the third table. Tables are available for download.

## 3. Materials and Methods

### 3.1. Workflow and Scoring Scheme of multiple UniReD

UniReD is a computational tool used to predict functional associations between proteins based on a machine learning algorithm called MCL [19]. The workflow and the scoring scheme of the new version of UniReD, called *multiple* UniReD, are summarized below and presented analytically in Figure 3.

The user uploads two lists in .csv or .tsv format. These are: (*i*) a reference list (of proteins that are known to be implicated in a specific disease or biological pathway) and (*ii*) a query list (list of proteins whose associations with the proteins of the reference list the user would like to identify). Proteins in each input list should be in comma- or tab-separated format using UniProt/SwissProt accession numbers.

The user must select the relevant species to be analyzed (currently *Homo sapiens* or *Mus musculus*).

*multiple* UniReD searches for the existence of any association in UniReD clusters between the protein pairs of the two lists (reference and query list). We followed the same scoring scheme as in [22,23]. If the pair is present in a UniReD cluster, the association will obtain a score of 1 and UniReD will continue to the next pair of reference and query proteins. If a protein has not been analyzed by UniReD, no results will be retrieved. These proteins will be reported in a separate table upon analysis completion. 

If *multiple* UniReD cannot find any association within the clusters, it will proceed to the next step in order to search for paralogues. More specifically, *multiple* UniReD will search for a co-occurrence of a paralogue of the query protein with the reference protein in a UniReD cluster. If such a pair is present, the association will obtain a score of 0.5.

In the case that UniReD does not predict any association within a query paralogue and the reference protein, then *multiple* UniReD will investigate whether the query protein is part of a complex. In this case, it will search for co-occurrences of the reference protein in the complex. If such an association is confirmed, the association will obtain a score of 0.5.

Finally, if none of the above applies, UniReD will search for a relation between the orthologues of reference and query proteins. If such an association is documented, it will obtain a score of 0.5.

When none of the aforementioned cases are applicable, then no score will be assigned to the specific protein pair.

If, in any of the aforementioned steps, a score is assigned to a protein pair, the procedure continues to the next protein pair without searching for any further association (demonstrated in Figure 3). The overall score for each query protein is calculated as the sum of the scores of the query protein with each of the reference ones. The results are returned as a table in a descending order by the query protein’s overall score, and can be downloaded in a comma/tab-separated format. 

### 3.2. Data Integration from Various Resources Used by multiple UniReD

*multiple* UniReD uses protein clusters and PPIs predicted by UniReD version October 2021 using an inflation value of 2.0 for the creation of protein clusters. For each protein found in any cluster formed by UniReD’s methodology, we integrated information regarding paralogues, human–mouse orthologues and complexes. Paralogue protein pairs were obtained from Ensembl [24] for both *H. sapiens* and *M. musculus* species (data retrieved on 25 May 2022 (*H. sapiens*) and 21 June 2022 (*M. musculus*)). Orthologues were also obtained from the same source on 25 May 2022. Complex information originates from the ComplexPortal [25] resource at EBI (data retrieved on 25 May 2022 (*H. sapiens*) and 21 June 2022 (*M. musculus*)). Furthermore, gene names, NCBI gene names and Ensembl gene IDs were retrieved from UniProt and Ensembl (data retrieved on 25 May 2022 (*H. sapiens*) and 21 June 2022 (*M. musculus*)). The result of this procedure was a dictionary (JSON format) summarizing information for each protein found to appear in clusters by UniReD. Each protein entry in this dictionary is represented by the UniProt/SwissProt accession number of proteins. For each one of them, fields for the gene name, NCBI gene name, Ensembl gene ID, paralogues, ComplexPortal complex id and mouse–human orthologues are provided. Finally, labels are used to highlight the existence of reviewed mappings between a UniProt accession number and all aforementioned identifiers.

### 3.3. Implementation and Running Time

The pipeline was developed in a Python 3 environment [25] (Python 3.7). The web interface was built using R and Shiny. The reticulate package was used in order to integrate the Python script with R [26] (R Core Team, 2022) and Shiny [27] (Winston et al., 2021). Regarding runtime, it takes just a few seconds for an average reference and query list of 100 proteins.

### 3.4. Statistical Analysis of the Results

We produced 1000 random lists of query proteins, and we calculated the sum of the overall score with the same list of reference proteins. The score differs significantly when compared to the sum of the overall score of the test case (*p* value < 10^−3^).

## 4. Conclusions

There are several experimental or in silico approaches to confront the important problem of PPI discovery/prediction. Living in the -omics era, there are different large-scale experimental approaches which produce a lot of results that can sometimes be confusing and difficult to interpret. Therefore, emerging computationally predicted PPIs can provide strong indications and hints about putative PPIs and thus complement the wet lab findings to maximum exploitation. UniReD is a tool that, among other things, predicts PPIs using biomedical literature mining and, with the updated feature of *multiple* UniReD, assists in the identification/validation of putative biomarkers. It accepts two lists of genes, one with genes known to participate in a specific pathway/disease and a second one with genes suspected to be involved in this particular pathway/disease. UniReD ranks the latter in relation to the first one, thus helping in steering the experimental verification, which is required for the confirmation of the PPI prediction, in the right direction. Biomarker validation can be of great importance, because, amongst others, can shed light on disease biology by revealing new unknown interactions, unveil potential novel therapeutic targets and contribute to the understanding of the pathogenic processes.

## Figures and Tables

**Figure 1 ijms-23-11112-f001:**
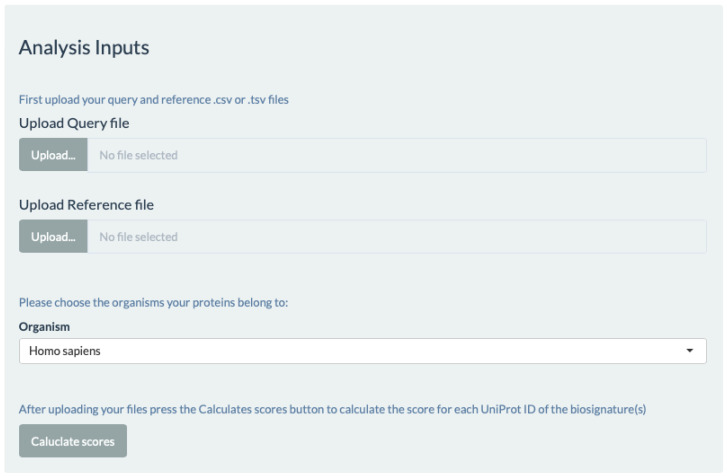
The input files required for the analysis using the web interface. The query input requires UniProt accession IDs for the proteins under investigation. The reference input requires UniProt accession IDs as well for the proteins known to be implicated in a specific biological process or disease. The organism input refers to the organism that the UniProt accession IDs belong to. The “Calculate scores” button calculates the scores for each of the UniProt accession IDs or the query.

**Figure 2 ijms-23-11112-f002:**
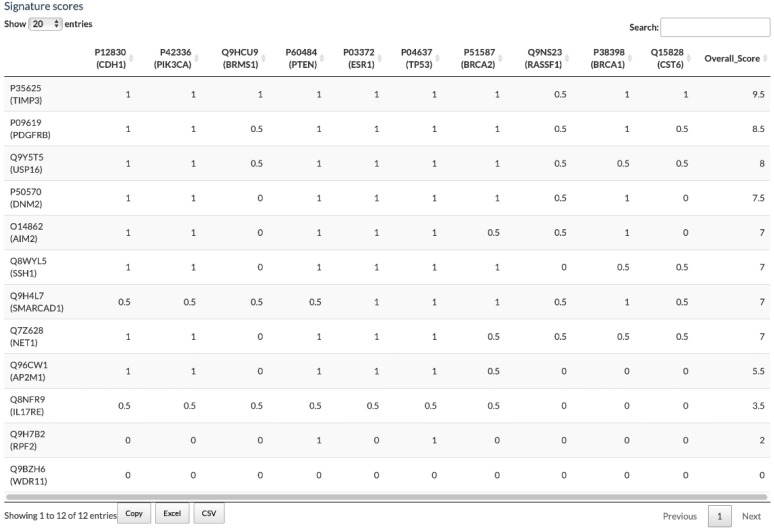
*multiple* UniReD output of the breast cancer biomarker analysis. The scoring table for each UniProt accession ID. The rows correspond to the query UniProt accession IDs whereas the columns to the reference UniProt accession IDs. Inside the parentheses is the corresponding gene name. The last column contains the overall (sum) score for each of the query UniProt accession IDs.

**Figure 3 ijms-23-11112-f003:**
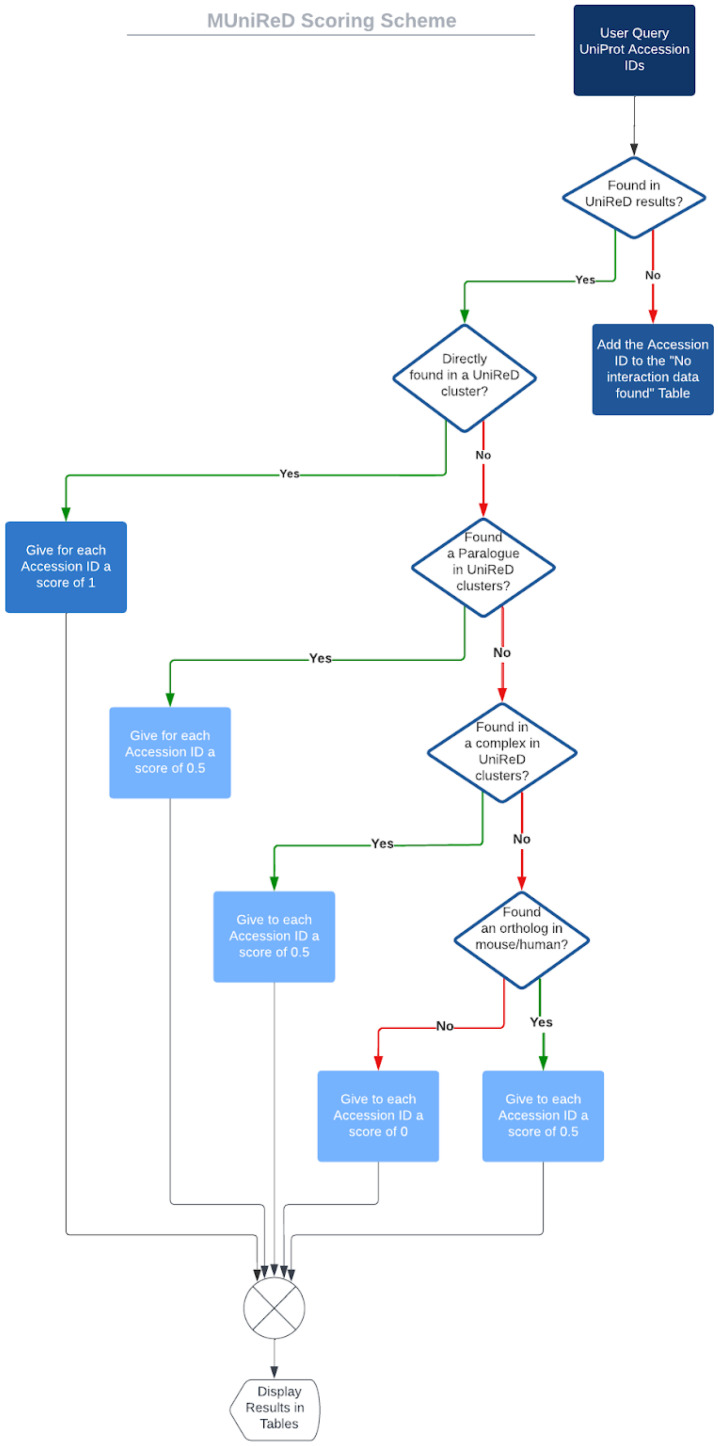
Workflow and scoring scheme of *multiple* UniReD.

## Data Availability

All data analyzed in this study are publicly available.

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
