# Peer review of "Prediction and Ranking of Biomarkers Using multiple UniReD"

_ijms, 2022, doi:10.3390/ijms231911112_

Round 1

Reviewer 1 Report

The authors introduced a new feature “multiple UniRed” to the original UniRed website, and claimed it allows users to identify potential protein biomarkers and their implications in disease by uncovering the associations between proteins in the reference and query lists using a text-mining algorithm called Markov Cluster.

However, the current study has very limited novelty and significance added to their original work (Theodosiou et al., 2020), and the scoring scheme applied to the system is lack of justification. For calculating the overall scores, each protein-protein interaction should be weighted by the level of evidence supported by the literature, and the proteins compiled from the high-throughput experiments can be ranked based on their fold changes or p-values. In addition, no comparison between their tool and others was provided, and the method has lots of room for improvement.

Author Response

We thank the reviewer for the comments. We followed the same scoring scheme that was used in previous publications.

This information was not included in the first draft of the article but now we added it (line 115). It is an arbitrary scoring system that is used for the ranking of the proteins of interest in order for the user to get an outline of the results. The user, of course, can explore the results and inspect the details of the analysis. As far as the comparison with other tools is concerned, to the best of our knowledge we are not aware of a tool conducting a similar kind of analysis (accepting two lists, a reference one and a query one and ranking the latter).

Reviewer 2 Report

Baltsavia et al. described  an update enhancing the existing PPI prediction tool UniReD. Their text and workflow are very clear and they demonstrate, using a use case, that it is working (.i.e. results can be retrieved). 

Here following my questions and comments:

  • The nature of the update should be clarified. The introduction is explaining the update like this "The whole process was performed manually and required the provision of two lists of genes/proteins( …) Herein we present a new powerful feature of UniReD tool (multiple UniReD) which automates the aforementioned process, previously conducted in a laborious way. " As the older version used also 2 lists, I didnt see at that point what was previously manual. Then as the material and methods describe the update as the automation of the different steps (direct interaction, to orthologs), it seemed that the new automated part refers to those steps. However, when testing the tool, it was clear that the original version only search for one protein at a time.
  • As the tool is explained to be useful for biomarker discovery, it would be important to have both a negative and a positive control, to give an idea of what the tool can give as output in both cases. The negative control can either proteins known not to be associated, or random protein lists.
  • As a positive control, the use case can be used, however a positive control using proteins very well known to interact in a disease would be even more strongly convincing. 
  • For the testing of the tool by new users, it would be an asset to propose 2 sample protein lists in it (eg the lists used for the positive control)
  • Is there a also way to interrogate the database programmatically? 
  • Are the parameters of the query shown somewhere and can be retrieved  for reporting?
  • Figure 3 should show the global score.
  • I think the title should be modified to be more consistent with the results, as there is no experimental part on biomarker validation. The tool can be used to confirm that a potential biomarker is indeed associated with proteins from a disease, but downstream experiments ( independent datasets ) are required to validate it as a biomarker.

Author Response

  • The nature of the update should be clarified. The introduction is explaining the update like this "The whole process was performed manually and required the provision of two lists of genes/proteins( …) Herein we present a new powerful feature of UniReD tool (multiple UniReD) which automates the aforementioned process, previously conducted in a laborious way. " As the older version used also 2 lists, I didnt see at that point what was previously manual. Then as the material and methods describe the update as the automation of the different steps (direct interaction, to orthologs), it seemed that the new automated part refers to those steps. However, when testing the tool, it was clear that the original version only search for one protein at a time.

We thank the reviewer for the comment. We have modified the introduction accordingly and we hope that we have clarified this particular point (last two paragraphs of the introduction).

  • As the tool is explained to be useful for biomarker discovery, it would be important to have both a negative and a positive control, to give an idea of what the tool can give as output in both cases. The negative control can either proteins known not to be associated, or random protein lists.

We thank the reviewer for this suggestion. We conducted an analysis based on random protein lists and the results are included in the article in the methods section (2.4).

  • As a positive control, the use case can be used, however a positive control using proteins very well known to interact in a disease would be even more strongly convincing. 

We thank the reviewer for the comment. The reference list that we used in the test case contains proteins that are known to implicate in the specific disease. We modified the text accordingly to clarify this matter (lines 194-197)

  • For the testing of the tool by new users, it would be an asset to propose 2 sample protein lists in it (eg the lists used for the positive control)

We thank the reviewer for the comment. In the help page we included the two lists as the reviewer reccomended “If you want to run an example you can use this query file and this reference list file.”

  • Is there a also way to interrogate the database programmatically? 

In the current version of multiple UniReD this is not applicable but in the future updates we should definitely implement this option.

  • Are the parameters of the query shown somewhere and can be retrieved  for reporting?

We thank the reviewer for this remark. We modified the text accordingly (lines 143-14). The user can only choose the species to be analyzed.

  • Figure 3 should show the global score.

We modified the figure accordingly.

  • I think the title should be modified to be more consistent with the results, as there is no experimental part on biomarker validation. The tool can be used to confirm that a potential biomarker is indeed associated with proteins from a disease, but downstream experiments ( independent datasets ) are required to validate it as a biomarker.

We thank the reviewer for the comment. Indeed the experimental validation is required so we modified the title.

Reviewer 3 Report

In the manuscript, the authors showed the scoring scheme and the workflow of multiple UniReD tool to obtain the potential functional associations between proteins-of-interest. Such computational tool is developed upon the published bioinformatic tool, UniReD using a machine learning algorithm, MCL that has been validated for the prediction and discovery of new protein-protein interactions. However, whether the current multiple UniReD is also valid for multiple protein-protein interactions is poorly supported by any experimental data or rationales thereby the readers in the journal might not feel confident to use multiple UniReD version. Therefore, there are several major questions to be answered before the acceptance of the paper. 

1. The authors simply validated the multiple UniReD tool by using the tested data from the previous work. Is it applicable to the other functional genes. Also, the authors claimed that the results of the analysis are slightly different and gave a reason for that. However, it is truly hard to follow up such demonstration. The authors can have a figure for the elucidation. 

2. Although there have been the experimental validation of UniReD for protein-protein interactions, the updated version of multiple UniReD still needs experimental validations using the other different protein-protein interaction. 

3. The authors need to make a comparison to the other tools which can conduct multiple PPI prediction. 

4. In figure 3, the authors need to provide a detailed discussion about the scoring number and the associated proteins. Also, since it is multiple UniReD, it will be more feasible to have protein names as the output in the table if possible. 

Author Response

  1. The authors simply validated the multiple UniReD tool by using the tested data from the previous work. Is it applicable to the other functional genes. Also, the authors claimed that the results of the analysis are slightly different and gave a reason for that. However, it is truly hard to follow up such demonstration. The authors can have a figure for the elucidation. 

multiple UniRreD is applicable to other functional genes assuming that the user will provide the reference and the query list. As far as the test case is concerned, we modified the text accordingly “because the two versions of the Uniprot database vary and this difference has an impact on UniReD results”(line 203).

  1. Although there have been the experimental validation of UniReD for protein-protein interactions, the updated version of multiple UniReD still needs experimental validations using the other different protein-protein interaction. 

We thank the reviewer for the comment. UniReD is a computational tool for predicting protein-protein interactions and functional associations. An experimental validation will prove whether the prediction is valid. In order to emphasize this fact, we included in the conclusions the following statement “which is required for the confirmation of the PPI prediction”

  1. The authors need to make a comparison to the other tools which can conduct multiple PPI prediction. 

To the best of our knowledge we are not aware of a tool conducting a similar kind of analysis (accepting two lists, a reference one and a query one and ranks the latter).

  1. In figure 3, the authors need to provide a detailed discussion about the scoring number and the associated proteins. Also, since it is multiple UniReD, it will be more feasible to have protein names as the output in the table if possible. 

We thank the reviewer for the comment. We have modified the figure accordingly (gene names shown now under the Uniprot Accesion numbers and not next to them. There is an extra table provided by the analysis which is not shown here but it is described in detail in the text under the figure 3 legend. If the reviewer means the scoring scheme in general this is described in Materials and methods section.

Round 2

Reviewer 1 Report

The authors did not proactively resolve the issues and concerns raised in my previous comments, unfortunately, the same problems persist.